# Clinical Insights into Mesenchymal Stem Cell Applications for Spinal Cord Injury

**DOI:** 10.3390/ijms262412139

**Published:** 2025-12-17

**Authors:** Matthew Shkap, Daria Namestnikova, Elvira Cherkashova, Daria Chudakova, Arthur Biktimirov, Konstantin Yarygin, Vladimir Baklaushev

**Affiliations:** 1Federal Center of Brain Research and Neurotechnologies of the Federal Medical and Biological Agency of Russia, 117513 Moscow, Russia; shkap010@mail.ru (M.S.); tchere@yandex.ru (E.C.); biartur2006@yandex.ru (A.B.); baklaushev@fccps.ru (V.B.); 2Center for Precision Genetic Technologies for Medicine, Engelhardt Institute of Molecular Biology of the Russian Academy of Sciences, 119991 Moscow, Russia; 3Department of Neurology, Neurosurgery and Medical Genetics, Department of Medical Nanobiotechnology, Pirogov Russian National Research Medical University of the Ministry of Healthcare of the Russian Federation, 117997 Moscow, Russia; 4Orekhovich Research Institute of Biomedical Chemistry of the Russian Academy of Sciences, 119435 Moscow, Russia; kyarygin@yandex.ru; 5Russian Medical Academy of Continuous Professional Education, 125284 Moscow, Russia; 6Federal Scientific and Clinical Center for Specialized Types of Medical Care and Medical Technologies of the Federal Medical and Biological Agency of Russia, 115682 Moscow, Russia

**Keywords:** cell therapy, mesenchymal stem cells, MSCs, spinal cord injury, SCI, clinical trials

## Abstract

This review examines the safety and clinical efficacy of mesenchymal stem/stromal cells (MSCs)-based therapies in patients with spinal cord injury (SCI). The analysis covers 26 clinical studies conducted on patients with varying degrees of the post-SCI neurological deficit. The review highlights the methodology of trials, the source of MSCs, the dosage of cells administered, transplantation methods, patient inclusion criteria, and the methods of evaluating the effectiveness of the therapy. MSC transplantation in SCI was safe and feasible in all the studies summarized in our review. All studies conducted have demonstrated varying degrees of patient improvement and reduction in the severity of neurological deficits. However, further controlled randomized studies on larger numbers of patients are needed to better evaluate the therapeutic efficacy of MS transplantation. The prospects of the enhancement of the efficacy of the SCI cell therapy with MSCs, including their transplantation with other types of stem cells, administration of MSC-derived exosomes, genetic modification of MSCs, use of the MSC- and other-stem-cell-based tissue-engineered scaffolds, and combination of cell therapy with neuromodulation, are discussed.

## 1. Introduction

Currently, the epidemiological significance of spinal cord injury (SCI) is increasing worldwide and leads to permanent lifelong motor, sensory and autonomic nervous system disorders, significantly impairing the quality of life and limiting the social and professional activity of patients [1]. The global incidence of SCI is 26.48 cases per million people per year, with a predominance in the male population (the male-to-female ratio is 3.2:1) [2,3]. In the Russian Federation, approximately 8000 patients with varying severity of SCI (American Spinal Injury Association ASIA) Impairment Scale (AIS) A-D) are registered annually [4,5]. SCI leads to permanent neurological deficit as a result of both primary mechanical damage and secondary damage of the spinal cord caused by neuroinflammation and oxidative stress [6]. Modern approaches of SCI treatment include fracture reduction, surgical decompression of the spinal canal, spinal stabilization, and neurorehabilitation, which aim to reduce secondary damage, but are not capable of significantly enhancing neuroregeneration [1,7]. The economic impact of SCI is substantial: in the absence of effective treatment, the primary expenses are driven by prolonged rehabilitation, which in many cases, fails to restore work capacity in individuals with severe (AIS A) or moderate (AIS B–C) SCI, thereby imposing a considerable burden on healthcare systems [8,9]. Due to the high incidence and disability rates among patients, it is important to develop new approaches to SCI therapy, one of which could be stem-cell-based therapy [7].

A wide range of stem cell types have been investigated in preclinical and early-stage clinical studies for the treatment of experimental spinal cord injury in animal models and humans, including mesenchymal stem/stromal cells (MSCs), neural stem and progenitor cells, glial progenitor cells, olfactory ensheathing cells, Schwann cells, bone marrow mononuclear cells, hematopoietic stem cells, and others [10,11,12]. Among all cell types, MSCs are the most studied candidates for clinical application [13,14].

MSCs are multipotent cells which can differentiate into mesodermal lineages (osteoblasts, adipocytes, chondrocytes, and others), and can be relatively easily isolated from bone marrow, adipose tissue, and other sources, including human placental Wharton’s jelly, without any notable ethical concerns [15]. Fundamental in vitro and in vivo studies in recent years have expanded our understanding of the mechanisms of MSC-mediated therapeutic effects in SCI models. The ability of MSCs to replace lost cells in central nervous system (CNS) diseases has not been clearly demonstrated in any fundamental study [16]. The predominant pro-regenerative mechanism of action of MSCs is via paracrine signaling modulating macrophage and microglial polarization at injury sites, activation of regulatory T cells, and extracellular matrix remodeling. The MSCs secretome comprises a range of factors with immunomodulating properties, for example, interleukin 8, interleukin 10 and others (reviewed in [17]). MSC-derived exosomes carry diverse bioactive molecules exerting reparative, immunomodulatory, angiogenic, anti-apoptotic, anti-ferroptotic effects and other pro-regenerative effects. The molecular cargo of such exosomes includes a large number of proteins, including but not limited to several neurotrophic factors, namely, nerve growth factor (NGF), glial cell line-derived neurotrophic factor (GDNF), and brain-derived neurotrophic factor (BDNF) [18]. Furthermore, it contains an array of RNAs, including messenger RNAs [19] and regulatory non-coding RNAs such as mature microRNAs or their precursors [20], long non-coding RNAs [21] and circular RNAs [22]. Additionally, lipids [23] and N-glycans [24] are found in MSC cargo. Remarkably, it has been proposed that MSC-derived exosomes might also carry mitochondria [25,26], which is still a topic of debate. Given space constraints, we refer readers to a recent comprehensive review on this subject [27].

In addition to their paracrine action, MSCs can affect surrounding cells through direct contact, for example, by transferring mitochondria (through nanotunnels, cell fusion, gap junctions, microvesicles, or isolated organelles). This mitochondrial transfer may exert a neuroprotective effect in various pathological conditions by restoring cellular energy reserves and supporting cell viability [28,29,30]. It is also known that MSCs can mediate their effects by interacting with cells in their microenvironment, such as endothelial cells [16,31,32]. The potential of MSCs as a therapeutic agent is further underscored by evidence that they can survive for extended periods of time (at least a week) following direct administration into the site of SCI, potentially exerting therapeutic effects through the mechanisms outlined above [33]. Taken together with their proven safety profile, low immunogenicity, and the relative simplicity and cost-effectiveness of large-scale production, allogeneic MSCs represent a promising candidate for the treatment of diseases and injuries of CNS. To the best of our knowledge, MSCs exert their action via the aforementioned mechanisms and eventually get cleared from the tissue without permanently integrating or differentiating into neural cells within CNS, and no signal of tumorigenicity has been reported in clinical follow-up to date.

Encouraging results from numerous preclinical and fundamental studies, along with advances in understanding the underlying mechanisms of cell therapy, have led to the initiation of clinical trials assessing the impact of MSCs on the progression and outcomes of SCI in humans. A recent comprehensive review by Zeng C et al. [34] summarized the advantages of clinical applications of MSCs, including direct transplantation, tissue-engineering scaffolds and MSC-derived exosome therapies, and highlighted the main molecular mechanism underlining the therapeutic effects of MSCs in SCI.

Given the increasing number of studies combining MSCs with other therapies for SCI (bioscaffolds, functional electrical stimulation, etc.), it has become challenging to evaluate the effects and safety of MSCs as a monotherapy without confounding factors introduced by additional treatments. A more isolated assessment would provide clearer insights into the curative potential and limitations of MSCs. The aim of the present narrative review is to comprehensively summarize and critically analyze all published clinical studies from 2000 to the present, offering a unique and up-to-date synthesis focused exclusively on MSC monotherapy for SCI.

## 2. Materials and Methods

A literature search was performed in the following databases to identify relevant studies: in PubMed, Medline, Web of Science, Google Scholar, and other open-source databases in August 2025. The following keywords were used: “Spinal cord injury”, “SCI”, “Spinal Cord Trauma”, “mesenchymal stem/stromal cells”, “MSC”, “clinical trial(s)”, “paraplegia”, “cell therapy”, “stem cell therapy”, “cell transplantation”, “safety”, and “efficacy”, in different combinations. The search was restricted to full-text articles in English published in 2000–2025. The study workflow is shown in Figure 1.

As a result of the initial search, more than 100 articles were found. After removing duplicate articles and those in languages other than English (found due to keyword similarities across languages), a total of 90 articles underwent a second-tier evaluation performed by two reviewers independently. Reviewers resolved any disagreements by reaching consensus. Articles to be reviewed were selected based on inclusion/exclusion criteria. Inclusion criteria: studies on adult participants with neurological impairments caused by SCI, pilot studies, case series, non-randomized and randomized clinical trials, MSC monotherapy. Exclusion criteria: reviews, case reports, preclinical studies, combination therapy studies, studies without a published full protocol. A total of 26 studies were selected. The information gathered from the studies comprised study phase, study design (controlled or uncontrolled, blinded or non-blinded), patient number, MSC dosage, MSC origin, tissue damage level, timing of therapy initiation after trauma, treatment outcomes, side effects of transplantation, and duration of follow-up.

The quality of the studies was assessed by a standard Physiotherapy Evidence Database scale (PEDro), with established quality ratings: 0–3 score classified as “poor”, 4–5 as “fair”, 6–8 as “good”, and 9–10 as “excellent” [35]. Table 1 provides a summary of the reviewed articles.

## 3. Results of Clinical Studies

Interest in the use of cell therapy for the treatment of SCI arose at the turn of the 20th and 21st centuries, when many authors attempted to evaluate the safety and efficacy of bone marrow mononuclear cell therapy in pilot clinical studies: Park H.C. et al. [62], Callera F. et al. [63], Syková E. et al. [64], Yoon S.H. et al. [65], Deda H. et al. [66], Geffner L.F. et al. [67]. It is known that bone marrow mononuclear cells contain MSCs, but in relatively small quantities (0.001–0.03%) [68,69], so we did not include these studies in the analysis. The earliest identified study on MSC-derived monotherapy dates to 2009 [36]. Thus, this review encompasses 26 studies on MSC therapy for SCI from 2009 to 2025 (over 17 years), with a summary of the data presented in Table 1.

### 3.1. Design of Clinical Trials

Tests were conducted in various countries, including China [40,41,51,54], Egypt [42], Brazil [43,48,60], South Korea [37,39,44,45], Pakistan [46], Turkiye [61], Spain [47,49,50,52], Japan [53,58,59], India [36,38,55], Jordan [57], and the United States [56], indicating global interest in this issue. Most clinical trials were phase I and/or II studies and were designed to demonstrate the safety and tolerability of treatment, as well as to preliminarily assess the efficacy of MSC transplantation in SCI. The number of patients included in the trials ranged from 5 to 68 per cohort. Control groups—comprising patients who received either placebo or standard treatment without cell therapy—were incorporated in fewer than one-quarter of the studies [40,41,42,51,52,55]. These studies also adhered to randomized controlled trial designs in accordance with current clinical trial standards.

Patient inclusion criteria varied significantly between studies. In particular, the level of SCI varied, ranging from isolated cervical [40,45,53,58,59,60], thoracic [46,47,48,52,61], or lumbar [41] lesions to mixed lesions in a majority of studies. The predominance of studies including patients with cervical spine injuries is probably due to the greater clinical significance of restoring function in this location of injury.

Variability was also observed in the degree of neurological deficit of the included patients on the AIS [70]. For instance, the studies by Cheng H. et al. [41] and Vaquero J. et al. [47] enrolled only patients with complete injury (AIS A), while other studies included patients with varying degrees of injury. The time after injury at the start of therapy also varied significantly, from early (up to 1 month) to late recovery periods (after 6 months). The majority of analyzed studies focused on patients with subacute and chronic SCI, predominantly those with severe injury (AIS A-B). This likely reflects both logistical factors and the high prevalence of patients with chronic SCI complicated by disabling neurological deficits, positioning them as the primary target population for regenerative therapies. However, despite the general trend, there have recently been studies describing the effects of MSC in patients with acute SCI [53,55,58], as well as in patients with moderate-to-severe SCI (AIS C-D) [59].

This variation in patient selection criteria should be taken into account when comparing data on the effectiveness of cell transplantation in analyzing various studies, since it is known that the potential for the spontaneous recovery of neurological deficits varies significantly between complete and incomplete spinal cord injuries and also depends on the duration of the injury [71]. Several researchers have additionally highlighted the critical importance of an accurate initial assessment of spinal cord injury severity, as misinterpretation—particularly in studies involving patients with acute trauma—may lead to an overestimation of therapeutic effects attributed to the intervention [72,73].

In the analyzed studies, either allogeneic or autologous MSCs derived from various sources, including bone marrow, adipose tissue, and umbilical cord, were used. The cells were administered via intrathecal injections, direct intramedullary injection into the lesion site, and/or intravenous infusion, with MSC doses ranging from 1 × 10^6^ to 1 × 10^8^ cells (the data are shown in Table 1).

Safety and efficacy were evaluated in patients over a follow-up period ranging from 6 months to 5 years (see Table 1). This variability reflects different approaches to defining the time frame for possible patient recovery, encompassing early indicators of remyelination as well as long-term functional adaptation. The methods used to evaluate therapeutic efficacy—including primary and secondary endpoints and assessment techniques—also differed across studies. AIS was most commonly employed as the primary efficacy measure, aligning with international standards for neurological assessment in spinal cord injury [70,74].

Additional assessment tools included specific functional scales to quantify patients’ levels of independence in daily activities, notably the Barthel Index and the Spinal Cord Independence Measure III (SCIM III) [75]. Urodynamic evaluations were performed to objectively assess recovery of pelvic organ function, while neuroimaging modalities such as magnetic resonance imaging (MRI) were utilized to characterize structural changes. Along with standard neurological scales, a number of studies [41,45,52,53,58] used electrophysiological methods to assess the integrity of the conduction pathways. Therapeutic efficacy was assessed through a series of longitudinal control measurements of selected outcome variables, typically conducted at baseline (pre-intervention), and subsequently, at 3, 6 and 12 months post-treatment.

Notably, several research groups conducted a series of sequential clinical trials evaluating the same type of MSCs with various modifications to the transplantation protocol, including cell dosage, administration route, and frequency [37,45,47,48,49,50,60]. These studies progressively demonstrated that higher MSC doses, repeated transplantations, and combined administration methods resulted in enhanced therapeutic efficacy.

### 3.2. Safety of MSC Transplantation

All analyzed studies, regardless of their design, confirmed the safety of MSC transplantation. Across patients with varying injury levels, injury durations, and degrees of neurological deficit as assessed by the ASIA scale, reported adverse effects were predominantly mild—such as headache, nausea, and transient fever—or not reported at all. For example, in the largest (in terms of the number of subjects; 70 patients) randomized, controlled phase I/II study conducted at Cairo University [42], it was reported that no severe adverse events were observed over an 18-month follow-up period after intrathecal administration of autologous bone marrow-derived MSCs. Specifically, no inflammatory or infectious complications, intracranial hypertension, or carcinogenic effects were reported, underscoring the favorable safety profile of MSC transplantation in spinal cord injury. Similarly, a number of studies employing intrathecal, intramedullary, or combined ways of MSC administration reported no severe adverse effects, even during long-term follow-up (Table 1). In the study by Kaplan N. et al. [61], patients received MSCs sequentially through three different routes—intrathecal, intramuscular, and intravenous—administered four times over two months, and even this complex combined administration protocol demonstrated good tolerability and safety.

It is important to note that among all methods of administration routes, intramedullary transplantation (administration of MSCs directly into the area of SCI) was associated with the highest risk of complications in the form of local pain syndrome in the area of surgical intervention, increased sensory disturbances, muscle rigidity, and increased body temperature [41,47,57]. The authors of these studies attribute such adverse effects primarily to the neurosurgical procedure rather than to the transplanted stem cells themselves. Despite these moderate side effects, patients receiving intramedullary MSC transplantation demonstrated significant neurological improvements following this delivery method.

Two studies have reported serious complications in patients with SCI, such as bacterial pneumonia, meningitis [47], urinary tract infection, acute bronchitis, and even the death of one patient due to aspiration pneumonia [58]. However, in all cases, complications were deemed nonspecific by the investigators and predominantly ascribed to comorbid illness, chronic trauma, prolonged intensive care admission, or the heightened risk of nosocomial infection, rather than to MSC transplantation itself.

### 3.3. Clinical Efficacy of MSC Transplantation

#### 3.3.1. Timing-Dependent Outcomes

In the majority of analyzed studies, where MSC transplantation was performed within the first six months after SCI, improvements on the AIS grade were observed [36,53,54,55,58]. In a recent phase I trial by Koda M. et al. [58], patients in the early recovery period after SCI (within 3 weeks post-injury) underwent a single intravenous transplantation of 15 × 10^6^ MSCs. By the end of the observation period, significant improvements in sensory and motor function were reported, and 6 out of 10 patients exhibited improvement on the Frankel neurological scale (a modified analog of AIS [76]). Similarly, an earlier case series study by Honmou O. et al. [53] documented improvement in AIS A grade in 12 of 13 patients with intravenous MSC administration during the acute phase of SCI. In one study, a direct comparison of the impact of MSC transplantation in subacute versus chronic cases was performed [36]. While no improvements in AIS grade were detected in both groups, functional status—measured by the Barthel Index—improved in 5 of 14 subacute patients but in none of the chronic SCI group.

The superior therapeutic outcomes observed with early initiation of MSC administration could be most likely attributable to enhanced neuroplasticity during the acute phase of injury and the reduced presence of irreversible pathological changes, such as glial scar formation and Wallerian degeneration of axons within the descending pathways distal to an injury site [77,78,79]. This pattern mirrors findings after neurosurgical interventions, where spinal cord decompression performed early post-injury is associated with better functional recovery of patients with SCI [80,81]. Furthermore, numerous preclinical studies demonstrate that stem cell transplantation administered during the acute and subacute phases of SCI yields the most favorable neurological improvements in animal models [82].

However, the majority of analyzed studies have reported MSC transplantations performed during the chronic phase following SCI, with intervals extending up to 26 years post-trauma. Despite this delayed timing, patients demonstrated varying degrees of neurological recovery, including restoration of motor, sensory, and autonomic functions, including pelvic organ control, thereby supporting the potential for functional restoration even long after injury. For instance, a controlled clinical trial by Dai G. et al. [62] demonstrated that autologous bone marrow-derived MSC administration in chronic SCI resulted in improved sensory and motor scores on the AIS in 9 of 20 treated patients, and that that was a significantly better outcome than observed in the control cohort. This therapeutic benefit may be attributed to the relatively high local MSC dose (8 × 10^5^ cells) delivered into the subarachnoid space adjacent to the injury site. These findings align with recent systematic reviews and meta-analyses, confirming the efficacy of MSC transplantation in traumatic SCI [12,83,84].

Summarizing the current evidence regarding the timing of traumatic SCI MSC transplantation, it is evident that earlier start of treatment yields the most favorable outcomes. Nonetheless, further large-scale, randomized controlled trials are necessary to precisely define the optimal “therapeutic window” for MSC transplantation.

#### 3.3.2. Influence of Injury Severity

Patients with incomplete SCI exhibit a markedly greater capacity for spontaneous motor and sensory recovery within the first 6–12 months post-injury, attributed to the preservation of conduction pathways and neuroplastic mechanisms. In contrast, complete injuries (AIS grade A) demonstrate severely limited recovery potential due to extensive anatomical damage of the spinal cord [85]. The aforementioned patient inclusion criteria in the studies analyzed in this review varied significantly concerning the severity of SCI, classified on the AIS grade from A to D. In most studies involving cell transplantation in patients with complete SCI (AIS grade A), functional improvements were minimal and progressed slowly [44,48]. Nevertheless, in some cases [56,58], even in patients with ASIA grade A, partial recovery of neurological deficits was observed, particularly when intervention was administered early and involved high doses of MSCs.

The recent study by Hirota R. et al. [59] further supports these findings. It reports a clinical series involving patients with moderate-to-severe SCI classified as AIS grade C-D, who were enrolled in the experimental arm after completing a course of physical therapy and reaching a neurological plateau. Following intravenous administration of autologous bone marrow-derived MSCs, only one out of seven patients exhibited an improvement in AIS grade from C to D. Nevertheless, all patients demonstrated either maintained or enhanced functional status. To the best of our knowledge, this represents the first study specifically highlighting neurorehabilitation potential in patients with incomplete chronic SCI.

#### 3.3.3. Effect of MSC Type

Clinical studies have reported transplantation of both allogeneic and autologous MSCs obtained from different sources [86]. However, due to substantial heterogeneity in study designs, it remains inconclusive which MSC type confers the greatest therapeutic benefit in SCI treatment. In most of the studies, the transplanted MSCs were autologous bone marrow-derived cells, likely because these cells had been previously used in pilot studies. Additional reasons for the preference of bone marrow-derived MSCs include their extensive characterization, the relative ease of their extraction and processing, and their well-established safety profile. Furthermore, a meta-analysis by Liu et al. [87] showed that transplantation of MSCs from bone marrow in humans led to superior functional outcomes compared to MSCs derived from alternative sources. In the studies conducted by Bydon M. et al. [56] and Hur J et al. [44], MSCs derived from adipose tissue were transplanted, demonstrating promising therapeutic effects, although their use was limited to these two investigations. Autologous MSC transplantation is considered safe and ethically unproblematic; however, its application in acute trauma is constrained by the time required for MSC collection and preparation. During the acute phase, the administration of allogeneic MSCs from pre-established banks of standardized and thoroughly characterized cells obtained from selected donors could be more appropriate. Additionally, several studies indicate that the regenerative capacity of MSCs is influenced by the donor’s age, diminishing as a consequence of natural aging processes [88]. This further supports the rationale for utilizing MSCs sourced from established cell banks.

Several clinical studies conducted over the past five years by Albu S. et al. [52], Yang Y. et al. [54], Awidi A. et al. [57], and Kaplan N. et al. [61], which have also demonstrated a pronounced positive therapeutic benefit of umbilical cord MSCs. Notably, fundamental animal research involving direct comparisons of MSCs from various sources revealed that umbilical cord MSCs confer superior therapeutic effects [89,90]; however, the efficacy of umbilical cord MSCs in humans requires further validation through large-scale randomized clinical trials.

#### 3.3.4. Impact of Transplantation Parameters

In the analyzed studies, MSCs were delivered via various routes: intramedullary injection directly into the injury site, intrathecal administration, or intravenous infusion. Intrathecal delivery was the most common approach, utilized in 16 of 26 studies [36,37,38,42,44,46,47,49,50,51,52,54,56,57,60,61], likely due to its minimal invasiveness and the advantage of delivering cells directly into the cerebrospinal fluid, including around the lesion site. Intramedullary administration, employed in 11 studies (see Table 1) [37,38,40,41,43,45,47,48,55,57,60], enables targeted delivery of cells to the lesion but carries a higher risk of complications related to the surgical procedure itself, as noted earlier. It should be noted that a study by Awidi A. et al. demonstrated that combined intramedullary and intrathecal administration resulted in superior therapeutic efficacy compared to intrathecal delivery alone [39].

In the studies conducted by Ra J et al. [39], Honmou O. et al. [53], Koda M. et al. [58], and Hirota R et al. [59], MSCs were transplanted intravenously into patients with SCI. These investigations reported significant improvements in both sensory and motor functions, which may be attributed to the paracrine actions of MSCs—particularly their anti-inflammatory and neurotrophic effects—which are well-documented in systemic transplantation for neurological disorders [91,92,93].

Jug M. et al. [94] reported a clinical case of a young patient with acute SCI (3 months after trauma at the C3-C4 level with subtotal spinal cord injury) to investigate the biodistribution of MSCs following intravenous and intrathecal transplantation using radionuclide imaging techniques. In this study, autologous MSCs were initially administered intravenously at a dose of 15 × 10^7^ cells, with one-third labeled with a technetium Tc-99m, and their distribution was tracked over the subsequent 20 h. The study demonstrated predominant retention of MSCs in the reticuloendothelial system, lungs, and bone marrow, while their detectable amount at the site of spinal cord injury was insignificant.

Two weeks later, the same dose of MSCs was re-administered intrathecally using the same labeling protocol. Post-intrathecal administration, MSCs initially localized near the injection site and gradually migrated toward the injury site, with signals detectable along the entire spinal cord after 20 h. These findings led the authors to conclude that despite the ability of MSCs to actively migrate to the site of injury after intravenous transplantation, intrathecal administration is a more effective and perceptive method of MSC transplantation in the case of SCI. Nevertheless, intravenous delivery may elicit systemic therapeutic effects via organs of the reticuloendothelial system [95].

It is important to note that among the analyzed studies, both the administered cell dose and the frequency of administration varied considerably. While most studies implemented a single transplantation of MSCs, several trials adopted protocols involving repeated administration [37,38,46,47,49,50,54,57]. Currently, clinical data are insufficient to establish an optimal dosing regimen. However, preclinical investigations indicate that the regenerative effects of MSCs on spinal cord repair are dose-dependent and are potentiated by repeated transplantation, a phenomenon likely applicable to MSC therapies in humans [96].

An overview of the various clinical trial designs and their outcomes is schematically illustrated in Figure 2. A timeline of clinical studies on MSC monotherapy for SCI is shown in Figure 3, illustrating major advances in administration routes, transplantation techniques, and trial design over the past two decades summarizing the chronological development of MSC monotherapy trials conducted between 2009 to the present. Despite the numerous studies conducted to date, the precise mechanisms—and their sequential interplay—by which MSCs exert therapeutic effects in spinal cord injury remain insufficiently understood and require further fundamental investigation using animal models of SCI [18]. A more comprehensive elucidation of these mechanisms may help to identify optimal transplantation parameters and inform strategies to enhance the efficacy of MSC-based treatments in future clinical studies.

## 4. Quality of Clinical Trials

Of all analyzed studies, 54% were Pilot/Phase I series, 38% were Phase II/III, and the remaining 8% were Case series, indicating overall heterogeneity and predominance of early phase studies over more rigorous Phase II/III (Figure 4).

We used the PEDro scale as a suitable tool for the assessment of the methodological quality of the studies with varying study designs. Most studies (n = 19) had a low PEDro score below 5, 23% of studies (n = 6) had scores of 6–8, and 4% had score in a 9–10 range (n = 1) (Figure 3). The threshold of PEDro score of 6 is recommended to be used for systematic reviews as one of the inclusion criteria. The high number of studies with a PEDro score below 5 was due to the high number of pilot studies (without randomization, blinding and a control group) in our narrative review.

Randomization and blinding are a key characteristic assessed by the majority of the Clinical Trial Evaluation Systems. Across the 26 studies presented in this review, most studies (77%) were non-randomized pilot studies (Figure 3) with varying designs and methods of MSC administration. Although some pilot studies reviewed in our study report successes of the therapy, after detailed evaluation and the introduction of randomization, the effect of MSC therapy appears to not be that pronounced; the most prominent outcome was improved bladder sensitivity and function [52,55]. Of 26 studies analyzed, only one study was controlled and randomized (4%) and only five were controlled, randomized and blinded (19%) (Figure 3). Of all studies analyzed, only two with the highest PEDro scores, 8 and 9, indicating best quality were randomized and double blinded (Table 1).

Based on this, direct comparison of therapeutic efficacy across the studies described in the review is challenging due to significant variations in the results, which might be attributed to differences in the ethnic characteristics of patients and national standards of medical care. Further, the studies did not have a unified protocol and differed significantly in their design, including the presence or absence of control groups without cell therapy, patient inclusion criteria, sources and characterization standards of MSCs, dosing regimens and administration frequency, transplantation methods, and approaches for evaluating therapeutic efficacy. Patient inclusion criteria varied significantly between studies, reflecting the lack of a standardized approach to selecting candidates for cell therapy at present. Variability was also observed in the degree of neurological deficit of the included patients on the AIS. Such methodological heterogeneity poses challenges for direct comparison of results across studies.

## 5. Future Prospects

Clinical trials have demonstrated that MSC monotherapy is safe, though its therapeutic efficacy can be potentially enhanced by combining it with other strategies (Figure 5) [18]. For example, experimental studies in animal models of SCI have shown that MSC transplantation in combination with tissue-engineering scaffolds can improve treatment outcomes and promote more targeted and controlled axon regeneration, improve the microenvironment and increase the survival of transplanted cells [97,98,99,100]. Active research and development efforts are underway in this rapidly evolving field. A pilot clinical study by authors who previously reported a study of MSC monotherapy in humans demonstrated the safety and feasibility of intramedullary administration of human umbilical cord MSCs in combination with a collagen scaffold [101]. This approach facilitated axonal growth along collagen fibers and inhibited glial scar formation.

Despite considerable progress in the application of native MSCs for treating SCI, their therapeutic potential is often insufficient for complete restoration of neurological function. In this regard, the development of genetically modified MSCs is a promising strategy for enhancing their regenerative properties and improving the effectiveness of cell therapy. Genetic modification enables the targeted expression of therapeutically relevant factors such as BDNF, GDNF and NGF, among others [102]. For instance, transplantation of MSCs engineered to overexpress interleukin-10 significantly improved motor function in a mouse model of complete SCI compared to native MSCs, which can be attributed to enhanced anti-inflammatory, anti-apoptotic, and neuroprotective effects of MSCs [103]. Similarly, Yang et al. [104] demonstrated that MSCs overexpressing neuropeptide S yielded superior therapeutic outcomes in a rat model of SCI, including improved motor recovery, marked reduction in scarring, increased neuroregeneration and neurogenesis. Furthermore, Jiang et al. [105] reported that erythropoietin-overexpressing MSCs exerted potent anti-apoptotic and neuroprotective effects in models of ischemic–hypoxic injury.

Another promising therapeutic strategy involves the administration of MSC-derived exosomes, which have demonstrated significant efficacy in treating SCI in preclinical models [106]. Liu et al. [21] showed that intravenous administration of exosomes derived from MSCs engineered to overexpress the tectonic family member 2 gene significantly enhanced functional recovery in a murine SCI model, presumably through inhibition of neuronal apoptosis and suppression of inflammation and oxidative stress in the injured spinal cord. Another example of successful use of modified MSCs for SCI treatment is the work of Chen et al. [107], in which rats with SCI were intravenously administered exosomes isolated from miR-26a-modified MSCs, showing marked recovery of neurological deficits in the animals. The authors observed activation of neuroregeneration, neurogenesis, and reduced glial scar formation. These effects were attributed to the activation of the PTEN/AKT/mTOR signaling pathway, which may be one of the potential targets for enhancing neuroregeneration [108].

An additional promising approach is the combined transplantation of MSCs with other types of stem cells [109]. Early clinical trials have explored this approach in patients with SCI. In a pilot study conducted by Oraee-Yazdani et al. [110], 11 patients with subacute severe SCI underwent a single intrathecal transplantation of autologous Schwann cells together with bone marrow MSCs. The study confirmed the safety and feasibility of such a combined cell transplantation approach and demonstrated its initial effectiveness in the form of improved motor, sensory, and pelvic functions in patients. A recent preclinical study by Kim et al. [111] demonstrated the higher efficacy of co-administration of MSCs and neural progenitor cells compared to monotherapy, as evidenced by more pronounced functional recovery of animals, enhanced axonal neurogenesis, and motor neuron maturation. The results of this study also demonstrated that stepwise combined cell transplantation was more effective than monotherapy (even with repeated MSC administrations) and had a pronounced neuroregenerative effect. The authors attribute the therapeutic effect to the synergistic action of different stem cell types and improvement of the microenvironment during combined transplantation [112].

Moreover, recent investigations have highlighted the potentiation of MSC therapy effects by combining cell transplantation with neuromodulation techniques. These studies demonstrated that simultaneous stimulation of spared conductive pathways enhances the overall therapeutic impact of MSC-based therapy [113,114].

Recently, our team conducted preclinical studies on MSCs for traumatic SCI, demonstrating efficacy when delivered within a fibrin hydrogel and platelet-enriched plasma matrix, followed by invasive neuromodulation below the lesion and rigorous physical rehabilitation (data forthcoming, Russian patent, 26 November 2024: https://fips.ru/EGD/734a8ff9-32b8-47f7-8c70-9382e8c2d185). These findings supported regulatory approval for a phase I/II clinical trial (FCBRN-RM01-2023, https://grlsbase.ru/clinicaltrails/clintrail/14481 accessed on 10 October 2025). This work and related studies aim to establish MSCs as key immunomodulatory and regenerative agents within multimodal traumatic spinal injury therapies.

## 6. Conclusions

The collective findings from the analyzed clinical trials indicate that MSC transplantation in SCI is safe and feasible.

Preliminary efficacy data suggest improvements in neurological function, including restoration of motor, sensory, and autonomic nervous system functions. However, the pilot nature of many studies, lack of control groups in several cases, and heterogeneity in trial design require cautious interpretation of these results. Here, it should be acknowledged that the field of MSC therapy for SCI is highly heterogeneous and generally low in clinical strength. To confirm the therapeutic efficacy of MSCs in SCI and establish an optimal transplantation protocol, larger multicenter randomized controlled trials are necessary.

After detailed evaluation and the introduction of randomization, the effect of MSC therapy appears to not be that pronounced, with the most prominent outcome being improved bladder sensitivity and function. Future strategies to enhance cell therapy efficacy may include combining MSC transplantation with established treatments such as neuromodulation, neurorehabilitation, and surgical intervention for SCI, as well as adjunctive approaches like functional electrical stimulation, the administration of MSC-derived exosomes, co-implantation with tissue-engineered scaffolds, genetic modification of MSCs, and co-administration with other cell types to potentiate neuroregeneration.

## Figures and Tables

**Figure 1 ijms-26-12139-f001:**
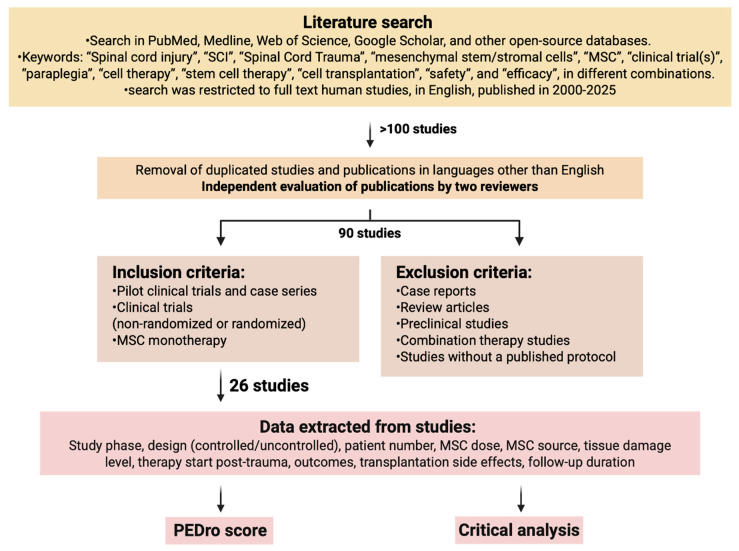
The study workflow. Data search strategies for clinical trials about MSCs in SCI (Created in BioRender. Shkap, M. (2025) https://BioRender.com/pkvhwko).

**Figure 2 ijms-26-12139-f002:**
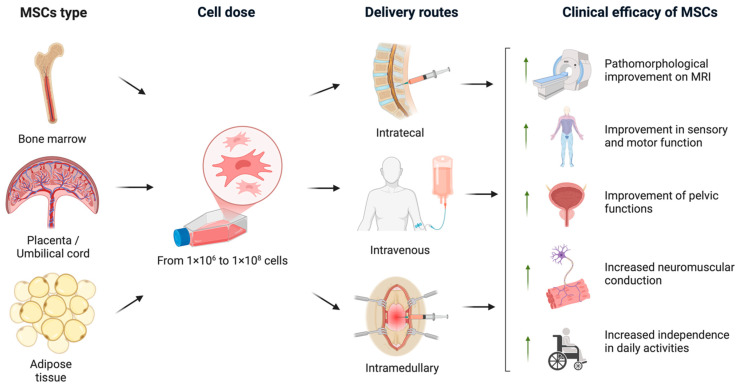
An overview of the various clinical trial protocols and the therapeutic efficacy of MSCs in SCI. In the analyzed clinical trials, MSCs were derived from bone marrow, adipose tissue, and placental or umbilical cord sources. The administered MSC dose ranged from 1 × 10^6^ to 1 × 10^8^ cells. The cells were transplanted via intrathecal, intravenous, or intramedullary routes. The most prominent clinical outcomes included pathomorphological improvements observed on MRI, enhanced sensory and motor function, recovery of pelvic organ function, improved neuromuscular conduction, and increased independence in daily activities (created in BioRender, Shkap, M. (2025) https://BioRender.com/uy6wvqo).

**Figure 3 ijms-26-12139-f003:**
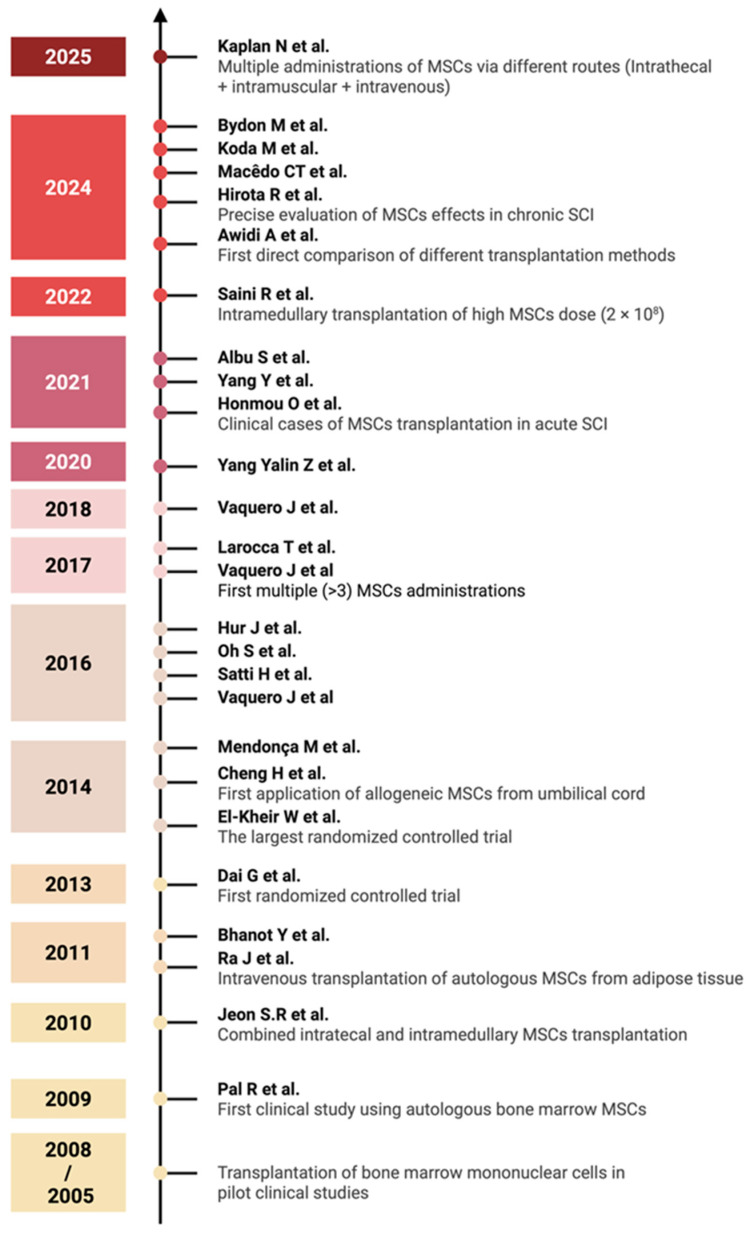
Timeline of clinical studies on MSC monotherapy for SCI, 2005–2025. Major milestones highlighted include the evolution of administration routes, the conduct of randomized controlled trials, and the development of transplantation methods [36,37,38,39,40,41,42,43,44,45,46,47,48,49,50,51,52,53,54,55,56,57,58,59,60,61]. Each year marks the publication of key studies that have influenced clinical practice in MSC-based SCI therapy (created in BioRender, Shkap, M. (2025) https://BioRender.com/zv7cnwh).

**Figure 4 ijms-26-12139-f004:**
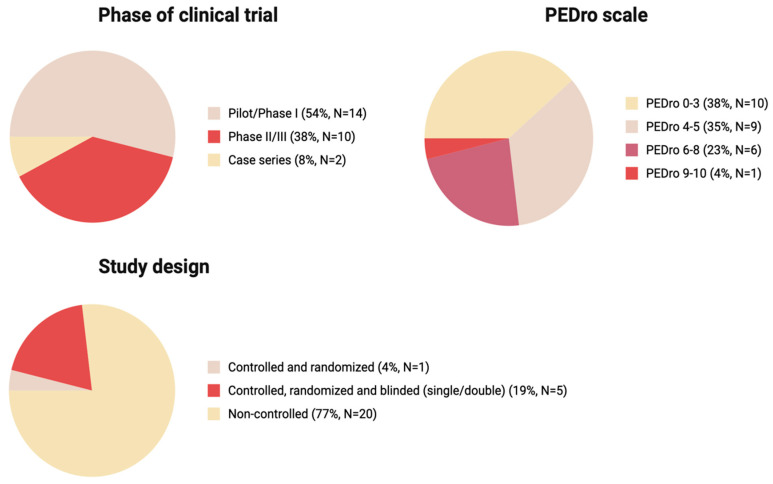
The key characteristics of the studies: phase of clinical trial, PEDro scale score, study design (Created in BioRender. Shkap, M. (2025) https://BioRender.com/35ozng7).

**Figure 5 ijms-26-12139-f005:**
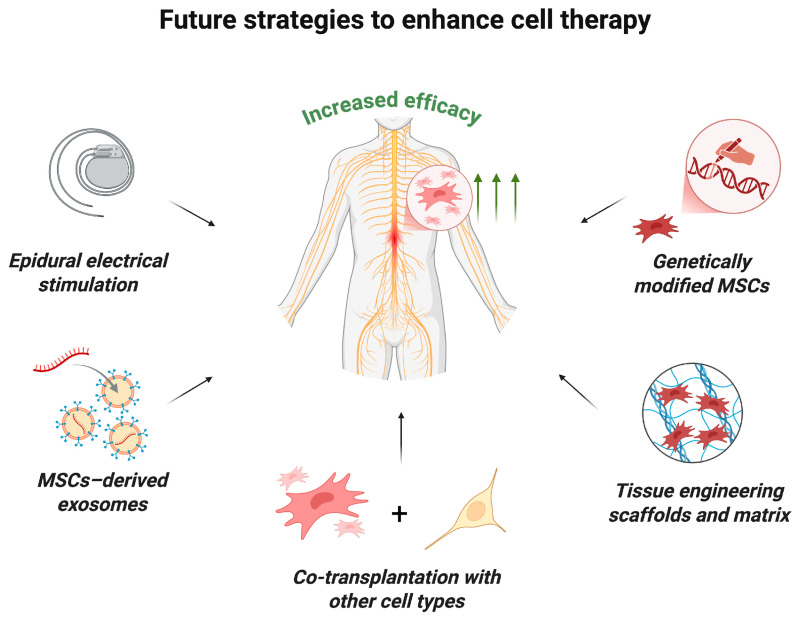
Future strategies to enhance MSC therapy. The efficacy of MSC transplantation may be increased by the following innovative approaches: the use of epidural electrical stimulation; combined transplantation with MSC-derived exosomes, other types of stem cells or tissue-engineering scaffolds and matrix; transplantation of genetically modified MSCs (created in BioRender, Shkap, M. (2025) https://BioRender.com/i3l4hes).

**Table 1 ijms-26-12139-t001:** Results of clinical studies included in the analysis on cell therapy for spinal cord injury.

Study	PEDro Scale	Phase/Control	Number of Patients/Age Range (Years)	Transplantation Route/ MSC Dose/Source	Level of Damage/Time to Start Therapy After Trauma/Follow-Up Period	Results	Transplantation-Related Side Effects
Pal, R. et al. [36] 2009 India	3	I/non-randomized, uncontrolled	30/18–55	Intrathecal single dose/1 × 10^6^ cells/kg/autologous bone marrow MSCs	Cervical, thoracic/Two groups of patients: 1–6 months (Group 1, n = 20) and more than 6 months (Group 2, n = 10)/12 months	Safety and feasibility confirmed. Significant improvement in urodynamic parameters. No improvement in AIS grade was observed.	Not identified
Jeon, S.R. et al. [37] 2010 South Korea	3	I/non-randomized, uncontrolled	10/34–61	Intramedullary/8 × 10^6^ cells/autologous bone marrow MSCs and then intrathecally twice/5 × 10^7^ cells/autologous bone marrow MSCs	Cervical, thoracic/1–108 months/12 months	Safety and feasibility confirmed. Significant improvement in sensitivity and motor function. No improvement in AIS grade was detected.	Not identified
Bhanot, Y. et al. [38] 2011 India	4	I/non-randomized, uncontrolled	13/18–60	Intramedullary once/1–4 × 10^6^ cells/kg/autologous bone marrow MSCs and then intrathecally twice/1–2 × 10^6^ cells/kg/autologous bone marrow MSCs	Cervical, thoracic/3–132 months/12 months	Safety and feasibility confirmed. Improvement in urodynamic parameters (n = 1). Improvement in sensitivity (n = 2). Improvement in AIS grade (n = 1).	Mild side effects associated with surgery
Ra, J. et al. [39] 2011 South Korea	3	I/non-randomized, uncontrolled	8/23–54	Intravenous single dose/4 × 10^8^ cells/autologous MSCs from adipose tissue	Cervical, thoracic/1.07–7.88 years/12 weeks	Safety and feasibility confirmed. Improvement in SCIM scores (n = 1). Improvement in AIS grade (n = 1).	Not identified
Dai, G. et al. [40] 2013 China	7	I, II/Randomized, single-blinded, controlled study	40 (20)/22–54	Intramedullary single dose/2 × 10^7^ cells/autologous bone marrow MSCs	Cervical/18–74 months/6 months	Safety and feasibility confirmed. Significant improvement in sensitivity, motor function, and urodynamic parameters compared to the control group. Improvement in AIS grade (n = 9).	Mild side effects associated with surgery
Cheng, H. et al. [41] 2014 China	6	II/Randomized, single-blinded, controlled trial	34 (10)/27–43	Intramedullary once/4 × 10^7^ cells/allogeneic MSCs from the umbilical cord	Lumbar/12–72 months/6 months	Safety and feasibility confirmed. Significant improvement in motor activity, muscle tone, Barthel index, and urodynamic parameters. No improvement in AIS grade was detected.	Mild side effect associated with surgery (radicular syndrome)
El-Kheir, W. et al. [42] 2014 Egypt	7	I, II/Randomized, single-blinded, controlled trial	70 (50)/16–45	Intrathecal once/2 × 10^6^ cells/kg/autologous bone marrow MSCs	Thoracic (n = 53), Cervical (n = 17)/12–36 months/18 months	Safety and feasibility confirmed. Significant improvement in sensitivity and motor function. Improvement in AIS grade (n = 17).	Not identified
Mendonça, M. et al. [43] 2014 Brazil	5	I/non-randomized, uncontrolled	14/18–65	Intramedullary single dose/5 × 10^6^ cells/cm3 of the volume of the irradiation zone/autologous bone marrow MSCs	Thoracic (n = 13), Lumbar (n = 1)/18–180 months/6 months	Safety and feasibility confirmed. Significant improvement in sensitivity and motor function, urodynamic parameters. Improvement in AIS grade (n = 7).	Mild side effects associated with spinal puncture (pain at the puncture site, 1 case of cerebrospinal fluid leakage)
Hur, J. et al. [44] 2016 South Korea	4	I/non-randomized, uncontrolled	14/20–66	Intrathecal single dose/9 × 10^7^ cells/autologous MSCs from adipose tissue	Cervical (n = 6), Cervicothoracic (n = 1), Thoracic (n = 6), Lumbar (n = 1)/3–28 months/8 months	Safety and feasibility confirmed. Improvement in sensitivity and motor function. No improvement in AIS grade was detected.	Mild side effects not associated with transplantation (urological infection, headache, nausea, vomiting)
Oh, S. et al. [45] 2016 South Korea	3	III (interim results of phase I/II published)/non-randomized, uncontrolled	16/18–65	Intramedullary once and subdurally once/1.6 × 10^7^ and 3.2 × 10^7^ cells/autologous bone marrow MSCs	Cervical/24–181 months/6 months	Safety and feasibility confirmed. Improvement in motor function. No improvement in AIS grade was detected.	Mild side effects (increased sensitivity disorders, muscle rigidity, pain syndrome)
Satti, H. et al. [46] 2016 Pakistan	4	I/non-randomized, uncontrolled	9/24–38	Intrathecally three times/1.2 × 10^6^ cells/kg/autologous bone marrow MSCs	Thoracic/2–55 months/9–27.5 months	Safety and feasibility confirmed. Efficacy not evaluated.	Mild nonspecific side effects (post-puncture headache, paresthesia)
Vaquero, J. et al. [47] 2016 Spain	5	I/non-randomized, uncontrolled	12/32–50	Intramedullary once and intrathecally twice/5–150 × 10^6^ (average 36 × 10^6^) and 30 × 10^6^ cells/autologous bone marrow MSCs	Thoracic/3–26 years/12 months	Safety and feasibility confirmed. Significant improvement in sensitivity and motor function. Improvement in AIS grade (n = 4).	Mild and moderate side effects (associated with surgical intervention)
Larocca, T. et al. [48] 2017 Brazil	4	I/non-randomized, uncontrolled	5/36–52	Intramedullary once/2 × 10^7^ cells/autologous bone marrow MSCs	Thoracic/25–111 months/6 months	Safety and feasibility confirmed. Improvement in superficial sensitivity. Improvement in AIS grade (n = 1).	Not identified
Vaquero, J. et al. [49] 2017 Spain	4	I/non-randomized, uncontrolled	10/34–59	Intrathecally four times/30 × 10^6^ cells/autologous bone marrow MSCs	Cervical (n = 5), Thoracic (n = 2), Lumbar (n = 3)/2.4–34.6 years/12 months	Safety and feasibility confirmed. Significant improvement in sensitivity and motor function, urodynamic parameters, reduction in neuropathic pain. No improvement in AIS grade was detected.	Mild side effects associated with spinal puncture (headache, pain at the injection site).
Vaquero, J. et al. [50] 2018 Spain	5	II/non-randomized, uncontrolled	11/28–62	Intrathecal three times/100 × 10^6^ cells/autologous bone marrow MSCs	Cervical (n = 4), Thoracic (n = 4), Lumbar (n = 3)/13.6 ± 14.79 years/10 months	Safety and feasibility confirmed. Significant improvement in sensitivity and motor function. Improvement in AIS grade (n = 3).	Not identified
Yang Yalin, Z. et al. [51] 2020 China	6	II/Randomized, controlled study	68 (34)/27–43	Intrathecal once/2 × 10^7^ cells/autologous bone marrow MSCs	Cervical (n = 44), Thoracic (n = 24)/15.4–25.9 months/6 months	Safety and feasibility confirmed. Significant improvement in sensitivity and motor function. No improvement in AIS grade was detected.	Mild side effects (back pain, headache, increased body temperature)
Albu, S. et al. [52] 2021 Spain	9	I, II/Randomized, double-blind, crossover, controlled	10/25–47	Intrathecal once/1 × 10^7^ cells/allogeneic MSCs from the umbilical cord	Thoracic/1–5 years/6 months	Safety and feasibility confirmed. Significant improvement in sensitivity. No improvement in AIS grade was detected.	Not identified
Honmou, O. et al. [53] 2021 Japan	-	Series of clinical cases	13/21–66	Intravenous single dose/0.84–1.6 × 10^8^ cells/autologous bone marrow MSCs	Cervical/43–54 days/6 months	Safety and feasibility confirmed. Improvement in sensitivity and motor function. Improvement in AIS grade (n = 12).	Not identified
Yang, Y. et al. [54] 2021 China	3	I, II/Non-randomized, uncontrolled	41/18–65	Intrathecal four times/1 × 10^6^ cells/kg/allogeneic MSCs from the umbilical cord	Cervical (n = 24), Thoracic (n = 7), Lumbar (n = 10)/More than 2 months/12 months	Safety and feasibility confirmed. Significant improvement in sensitivity and motor function. No data on improvement in AIS grade.	Mild side effects
Saini, R. et al. [55] 2022 India	8	I, II/Randomized, double-blind, controlled	27/18–50	Intramedullary once/2 × 10^8^ cells/autologous bone marrow MSCs	Cervical (n = 2), Lumbar (n = 25)/7–17 days/6 months	Safety and feasibility confirmed. Improvement in bladder sensitivity, postural control, and reduction in spasticity, but no improvement in motor function. Improvement in AIS grade (n = 6).	Mild side effects
Bydon, M. et al. [56] 2024 USA	4	I/Non-randomized, uncontrolled	10/18–65	Intrathecal once/1 × 10^7^ cells/autologous MSCs from adipose tissue	Cervical (n = 6), Thoracic (n = 4)/7–22 months/24 months	Safety and feasibility confirmed. Significant improvement in sensory and motor function. Improvement in AIS grade (n = 7).	Mild side effects
Awidi, A. et al. [57] 2024 Jordan	6	I, II/Non-randomized, uncontrolled	20/18–56	Group A: Intramedullary once/1 × 10^8^ cells/autologous bone marrow MSCs and then intrathecally three times/1 × 10^8^ cells/allogeneic MSCs from the umbilical cord Group B: Intrathecal three times/1 × 10^8^ cells/allogeneic MSCs from umbilical cord	Cervical (n = 7), Thoracic (n = 13)/14–228 months/1–3 years	Safety and feasibility confirmed. Improvement in sensory and motor function in both groups, with more pronounced improvements in Group A. Improvement in AIS grade in both groups (n = 16).	Mild side effects associated with intramedullary administration (local pain, headache, vomiting, fever)
Koda, M. et al. [58] 2024 Japan	3	I/Non-randomized, uncontrolled	10/27–67	Intravenous single dose/15 × 10^6^ cells/allogeneic bone marrow MSCs (Muse line)	Cervical/3 weeks/13 months	Improvement in sensory and motor function. Improvement in Frankel scale (n = 6).	Not identified as related to therapy
Hirota, R. et al. [59] 2024 Japan	-	Series of clinical cases	7/20–52	Intravenously once/1.00–1.90 × 10^8^ cells/autologous bone marrow MSCs	Cervical/1.3–27 years/6 months	Safety and feasibility confirmed. Significant improvement in sensory and motor function. Improvement in AIS grade (n = 1).	Not identified as related to therapy
Macêdo, C.T. et al. [60] 2024 Brazil	3	I/Non-randomized, uncontrolled	6/30–55	Intramedullary single dose/5 × 10^7^ cells/autologous bone marrow MSCs and then intrathecal single dose/5 × 10^7^ cells/autologous bone marrow MSCs	Cervical/24–192 months/12 months	Safety and feasibility confirmed. Significant improvement in sensory function. No improvement in AIS grade was detected.	Mild side effect associated with surgery
Kaplan, N. et al. [61] 2025 Turkey	3	I/Non-randomized, uncontrolled	6/19–39	Intrathecal + intramuscular + intravenous 4 times over 6 weeks/1 × 106 + 1 × 10^6^ + 1 × 10^6^ cells/allogeneic MSCs from the umbilical cord	Thoracic/6 months–12 years/12 months	Safety and feasibility confirmed. Significant improvement in motor and sensory functions, reduction in spasticity, improvement in pelvic functions. Improvement in AIS grade (n = 6).	Not identified

## Data Availability

No new data were created or analyzed in this study. Data sharing is not applicable to this article.

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
