# Peer review of "Clinical Insights into Mesenchymal Stem Cell Applications for Spinal Cord Injury"

_ijms, 2025, doi:10.3390/ijms262412139_

Round 1
Reviewer 1 Report
Comments and Suggestions for Authors
Overall, this is a well-written, comprehensive review of MSC transplantation for SCI. The focus of the review is on clinical trials. The review also highlights the challenges and potential suggestions to improve the therapeutic efficacy of MSCs. The following are minor comments.
Line 21-22 (Abstract). The way it is written reads like a research article. Change the word “evaluating” to “evaluation”.
Line 49: Modern therapeutic—the techniques described are not therapeutic. So, deleted the word “therapeutic”.
Table: Difficult to read, may format into Landscape format. If acceptable, reduce the font size to improve readability.
Line 221-222: Duplicated sentence
Author Response
Rebuttle Letter
Responses to Reviewer #1
Dear Reviewer,
Thank you very much for reviewing the manuscript and for your favorable evaluation of our work. We sincerely appreciate your insightful comments and have implemented the suggested revisions accordingly. We believe that the manuscript in current form meets the requirements for publication.
Reviewer: Overall, this is a well-written, comprehensive review of MSC transplantation for SCI. The focus of the review is on clinical trials. The review also highlights the challenges and potential suggestions to improve the therapeutic efficacy of MSCs. The following are minor comments.
Line 21-22 (Abstract). The way it is written reads like a research article. Change the word “evaluating” to “evaluation”.
Reply: Thank you for your suggestion, we have slightly changed the style of the abstract to make it more suitable for a review article.
Reviewer: Line 49: Modern therapeutic—the techniques described are not therapeutic. So, delete the word “therapeutic”.
Reply: Thank you for your comment. The correction has been made.
Reviewer: Table: Difficult to read, may format into Landscape format. If acceptable, reduce the font size to improve readability.
Reply: We fully agree with this comment. We originally created the table in Landscape format, but the format had changed upon upload. We wrote to the Editorial about this technical issue and believe that it will be fixed.
Reviewer: Line 221-222: Duplicated sentence.
Reply: Thank you very much for noticing. This is a typo, which we corrected.
Once again we thank you wholeheartedly for thorough review and valuable suggestions. We hope that in its current form the manuscript meets the criteria of MDPI for publishing.
Reviewer 2 Report
Comments and Suggestions for Authors
The manuscript presents a valuable and up-to-date overview of a clinically relevant topic. The subject is important and well supported by recent literature, but the paper would benefit from substantial revision before it can be considered for publication.
To make the work more solid and scientifically convincing, I would suggest the following improvements:
- Please clarify whether this is a narrativeor a systematic review. In my opinion, developing it into a proper systematic review would significantly strengthen the paper.
- Add a Methods section describing the literature search (MESH), and include a PRISMA flow diagram as well as a risk of bias assessment.
- Include a clearstatement of aim and novelty at the end of the Introduction, explaining what gap this review fills.
- In theDiscussion, provide a more critical comparison with previous reviews and add a short Study Limitations
- The Future Prospects section could be shortened and focused on the most promising directions.
- Abbreviations like AIS, SCI, and MSCs are introduced several times in different sections. They should be defined once when first mentioned and then used consistently throughout the text. This would make the manuscript easier to read and more professional.
Overall, this is a promising review, and with these revisions it could make a strong contribution to the field. Also, I recommend professional English editing to remove redundancies, shorten long sentences, and ensure clear and precise scientific writing. Unless authors are able to do find those mistakes themselves.
Comments on the Quality of English LanguageThe paper is written clearly overall, but the English needs some polishing before publication. There are a few grammatical mistakes, repetitions, and awkward sentences that make parts of the text harder to read. I would recommend a careful language check by a native speaker or professional editor to make the manuscript smoother and more consistent.
Author Response
Rebuttle Letter
Dear Reviewer,
Thank you very much for reviewing the manuscript and for your favorable evaluation of our work. We sincerely appreciate your insightful comments and have implemented the suggested revisions accordingly. We believe that the manuscript in current form meets the requirements for publication. Please find below our point-by-point response. All revisions in the updated manuscript are marked in blue.
Reviewer: The manuscript presents a valuable and up-to-date overview of a clinically relevant topic. The subject is important and well supported by recent literature, but the paper would benefit from substantial revision before it can be considered for publication.
To make the work more solid and scientifically convincing, I would suggest the following improvements:
1. Please clarify whether this is a narrative or a systematic review. In my opinion, developing it into a proper systematic review would significantly strengthen the paper.
Reply: Thank you for your comment.This is a narrative review, and we plan to keep it as such. In our opinion narrative reviews reach a broader audience beyond specialists and allow for a flexible, comprehensive discussion that better suits our paper’s goals. We will emphasize in the text that this is a narrative review.
2. Add a Methods section describing the literature search (MESH), and include a PRISMA flow diagram as well as a risk of bias assessment.
Reply: Thank you for your valuable suggestion. We have added a detailed description of our literature search strategy in the Introduction section. However, we chose not to include a PRISMA flow diagram to avoid redundancy and because another reviewer noted that parts of the manuscript are already “overly long and descriptive,” so we aim to keep the text concise. Risk of bias assessment is primarily a defining feature of systematic reviews, whereas our manuscript is a narrative review.
3. Include a clear statement of aim and novelty at the end of the Introduction, explaining what gap this review fills.
Reply: The aim of our review was to summarize data from all clinical studies utilizing MSC monotherapy for SCI over the past decades. Our review covers the majority of clinical trials conducted during this period to provide a comprehensive overview of the current state of this therapeutic approach and to outline future development prospects. The unique aspect of this work is that it presents a complete and up-to-date picture focused exclusively on MSC monotherapy. Monotherapy was chosen to specifically evaluate the effects and safety of MSCs alone, without confounding factors introduced by combined or adjunctive therapies, providing clearer insights into MSCs therapeutic potential and limitations.
4. In the Discussion, provide a more critical comparison with previous reviews and add a short Study Limitation
Reply: Thank you for this suggestion. In our view, the main limitation is as follows. Direct comparison of therapeutic efficacy across the studies described in the review is challenging due to variations in transplantation protocols, including MSC type, dosage, route of administration, timing, injury severity, patient age, and other factors.
5. The Future Prospects section could be shortened and focused on the most promising directions.
Reply: Thank you for your comments. In the Future Directions section we highlight novel technologies and approaches relevant to the review topic that have not been addressed in previously published reviews. This section also references studies by authors who have previously conducted randomized controlled trials involving MSCs (as detailed in the manuscript’s Table) but have since transitioned to combined therapies incorporating scaffolds and matrices. Retaining this section is essential, as it enhances the novelty of our work.
6. Abbreviations like AIS, SCI, and MSCs are introduced several times in different sections. They should be defined once when first mentioned and then used consistently throughout the text. This would make the manuscript easier to read and more professional.
Reply: We fully agree and appreciate your careful attention. We have addressed this issue consistently throughout the text.
7. Overall, this is a promising review, and with these revisions it could make a strong contribution to the field. Also, I recommend professional English editing to remove redundancies, shorten long sentences, and ensure clear and precise scientific writing. Unless authors are able to do find those mistakes themselves. The paper is written clearly overall, but the English needs some polishing before publication. There are a few grammatical mistakes, repetitions, and awkward sentences that make parts of the text harder to read. I would recommend a careful language check by a native speaker or professional editor to make the manuscript smoother and more consistent.
Reply: Thank you for your suggestion and for the recognition of our work’s merits. To enhance the manuscript's language quality, we have included a co-author who is a fluent English speaker with professional experience in scientific editing and substantial expertise in the subject matter.
We thank you wholeheartedly for thorough review and valuable suggestions. We hope that in its current form the manuscript meets the criteria of MDPI for publishing.
Reviewer 3 Report
Comments and Suggestions for Authors
This particular review offers a comprehensive summary of various clinical trials on monotherapy using mesenchymal stem cells in spinal cord injuries. The authors of the paper have discussed 26 trials from around the world. The paper appears to be quite educational as it deals with the latest research being carried out on the regeneration of spinal cords using stem cell therapy. Nevertheless, the review appears very descriptive without critical synthesis. The important aspects of comparison of the present meta-analysis with earlier ones, research methodology, and uniform assessment criteria seem inadequately discussed. Improvement in these domains may add increased scientificness as well as globalization aspects to the research paper.
Major Comments
1. The authors must define very briefly how it differs from existing ones in the same year (e.g., Biology 2023, 12(5)), 653. This needs a comparison stating new data, new years of data studied from 2009 to 2025, and new findings on outcomes.
2. The manuscript primarily deals with MSC monotherapy. A great deal of the literature reviewed contains a combination of various interventions. Please define your inclusion and exclusion criteria. Please highlight in your manuscript that your approach builds on existing analysis.
3. Table 1 is comprehensive but overly long and descriptive. Consider summarizing the quantitative outcomes (e.g., number/percentage of patients showing AIS improvement) in a synthesized table or figure to highlight overall efficacy trends.
4. While variability among studies is mentioned, the authors should analyze how specific differences (e.g., route of administration, acute vs chronic phase, dose, allogeneic vs autologous) influence clinical outcomes. A critical appraisal table or forest-style visualization would strengthen the analysis.
5. The review could improve by providing a more integrated section relating outcomes to mechanistic principles such as paracrine actions of MSCs or exosomes. This may be included as a schematic diagram.
6. The inclusion of 2024-2025 studies is commendable. However, please ensure completeness and cross-check against the clinicaltrials.gov database for any recently published or ongoing MSC-SCI trials not yet cited.
8. The assessment provides qualitative information on effectiveness but fails to assess the significance and relative weight of positive outcomes. Even a qualitative statistical analysis of significance (e.g. percentage trials with positive AIS grade outcomes) would be an upgrade.
9. Conclusions on safeties should be put into perspective by considering limitations: relatively short duration of follow-up, absence of standardized adverse events reporting, and untested tumorigenicity over long-term.
10. The authors interchange “SCI,” “TSM,” and “spinal trauma.” Please unify terminology throughout and define abbreviations clearly at first use.
11. Section 3 talks about future directions such as scaffolds, exosomes, and neuromodulation. Can you elaborate on the comparison of the outcomes of purely MSC monotherapy and these combinational therapies?
12. Adding a risk of bias assessment using either the Cochrane assessment or MINORS may improve the robustness of the review by highlighting methodological strengths and weaknesses.
Comments on the Quality of English LanguageThe manuscript requires English polishing to improve clarity and flow, particularly in transitions between subsections (e.g., between 2.1–2.4). Some sentences are lengthy and repetitive.
Author Response
Rebuttle Letter
Dear Reviewer,
Thank you very much for reviewing the manuscript and for your favorable evaluation of our work. We sincerely appreciate your insightful comments and have implemented the suggested revisions accordingly. We believe that the manuscript in current form meets the requirements for publication. Please find below our point-by-point response. All revisions in the updated manuscript are marked in blue.
Reviewer: This particular review offers a comprehensive summary of various clinical trials on monotherapy using mesenchymal stem cells in spinal cord injuries. The authors of the paper have discussed 26 trials from around the world. The paper appears to be quite educational as it deals with the latest research being carried out on the regeneration of spinal cords using stem cell therapy. Nevertheless, the review appears very descriptive without critical synthesis. The important aspects of comparison of the present meta-analysis with earlier ones, research methodology, and uniform assessment criteria seem inadequately discussed. Improvement in these domains may add increased scientificness as well as globalization aspects to the research paper.
Major Comments
1. The authors must define very briefly how it differs from existing ones in the same year (e.g., Biology 2023, 12(5)), 653. This needs a comparison stating new data, new years of data studied from 2009 to 2025, and new findings on outcomes.
Reply: The comprehensive review by Zeng CW, "Multipotent Mesenchymal Stem Cell-Based Therapies for Spinal Cord Injury: Current Progress and Future Prospects" (Biology, 2023 Apr 26;12(5):653), which we read promptly upon its publication in 2023, primarily focuses on the mechanisms of action of MSCs. Therefore it offers limited discussion on clinical trials involving MSCs, aside from briefly mentioning several clinical studies.
Following your suggestion, we have added a statement to the introduction highlighting the novelty of our work and the specific knowledge gap it addresses. Our review provides the most comprehensive and up-to-date summary of clinical studies involving MSC monotherapy for SCI, summarizing currently existing data on the subject. This fills a gap in knowledge by clarifying the current state of MSC monotherapy.
2. The manuscript primarily deals with MSC monotherapy. A great deal of the literature reviewed contains a combination of various interventions. Please define your inclusion and exclusion criteria. Please highlight in your manuscript that your approach builds on existing analysis.
Reply: Thank you for your comment. We have added the inclusion criteria and emphasized in the introduction the specific feature of our review, which is that it focuses exclusively on MSC monotherapy.
3. Table 1 is comprehensive but overly long and descriptive. Consider summarizing the quantitative outcomes (e.g., number/percentage of patients showing AIS improvement) in a synthesized table or figure to highlight overall efficacy trends.
Reply: The aim of our review was not to quantitatively assess the therapy effectiveness, as this has already been addressed in large systematic reviews and meta-analyses to which we refer. Our objective is to highlight new trends in MSC therapy and emphasize the effectiveness of combined approaches described in the "Future Directions" section. We also believe that information in the table is the very essence of this manuscript, thus we prefer to keep it as detailed as possible. Finally, in our opinion, adding yet another table or figure will make the text redundant and overly long.
4. While variability among studies is mentioned, the authors should analyze how specific differences (e.g., route of administration, acute vs chronic phase, dose, allogeneic vs autologous) influence clinical outcomes. A critical appraisal table or forest-style visualization would strengthen the analysis.
Reply: Our manuscript is a narrative review and critical appraisal is not a standard component of this review type. Nevertheless we agree that providing such info might have strengthened other types of review, and we also notice that previously published systematic reviews and meta-analyses do not provide such info.
5. The review could improve by providing a more integrated section relating outcomes to mechanistic principles such as paracrine actions of MSCs or exosomes. This may be included as a schematic diagram.
Reply: It is with great enthusiasm that we have followed the reviewer’s advice, as this subject represents another area of interest for our team. We have included several paragraphs dedicated to describing the mechanisms of action of MSCs and MSC-derived exosomes (such as paracrine effects, etc). We chose to present this information in text form because the manuscript already contains a table and an illustration. Furthermore, presenting it in text allows for easier citation of the relevant references.
6. The inclusion of 2024-2025 studies is commendable. However, please ensure completeness and cross-check against the clinicaltrials.gov database for any recently published or ongoing MSC-SCI trials not yet cited.
Reply: We wholeheartedly thank the reviewer for recognizing the merits of our work. While we have endeavored to incorporate the latest studies, to ensure the quality and reliability of the data presented in this review, we excluded studies registered on clinicaltrials.gov that lack peer-reviewed publications summarizing their results.
8.The assessment provides qualitative information on effectiveness but fails to assess the significance and relative weight of positive outcomes. Even a qualitative statistical analysis of significance (e.g. percentage trials with positive AIS grade outcomes) would be an upgrade.
Reply: We acknowledge that adding a quantitative statistical analysis, such as the percentage of trials with positive AIS grade outcomes, could provide additional insights. However, these measures would not be entirely objective due to the heterogeneity of the studies included in this review. Hence, we do not pool the studies together because they vary considerably in design, methodology, and patient populations, except for a subset of articles by the same authors. Therefore, a formal meta-analysis or aggregated statistical assessment seems not feasible in this context.
9. Conclusions on safeties should be put into perspective by considering limitations: relatively short duration of follow-up, absence of standardized adverse events reporting, and untested tumorigenicity over long-term.
Reply: As the reviewer stated themself in commentary 5, the main mode of action of MSCs is paracrine (secretion of bioactive molecules). The current view is that MSCs do not undergo pro-neuronal differentiation and do not integrate into the central nervous system, and various studies using different methods have shown that MSCs do not survive at the lesion site for a long period. They are eventually cleared from the tissue, which reduces the risk of adverse effects such as tumorigenicity. We will add a sentence about the absence of standardized adverse events reporting .
10. The authors interchange “SCI,” “TSM,” and “spinal trauma.” Please unify terminology throughout and define abbreviations clearly at first use.
Reply: Thank you for your comment. Done!
11. Section 3 talks about future directions such as scaffolds, exosomes, and neuromodulation. Can you elaborate on the comparison of the outcomes of purely MSC monotherapy and these combinational therapies?
Reply: It has been demonstrated in animal studies that combination therapy is more effective compared to monotherapy. We do not explore this topic in depth, since, to the best of our knowledge, there have been no similar meta-analyses in humans. Therefore, we do not discuss it in detail in the text.
12. Adding a risk of bias assessment using either the Cochrane assessment or MINORS may improve the robustness of the review by highlighting methodological strengths and weaknesses.
Risk of bias assessment is primarily a defining feature of systematic reviews, whereas our manuscript is a narrative review.
The manuscript requires English polishing to improve clarity and flow, particularly in transitions between subsections (e.g., between 2.1–2.4). Some sentences are lengthy and repetitive.
Reply: Thank you for your suggestion. To enhance the manuscript's language quality, we have included a co-author who is a fluent English speaker with professional experience in scientific editing and substantial expertise in the subject matter.
Once again we thank you wholeheartedly for thorough review and valuable suggestions. We hope that in its current form the manuscript meets the criteria of MDPI for publishing.
Reviewer 4 Report
Comments and Suggestions for Authors
This manuscript presents a comprehensive and well-structured review of clinical studies evaluating the safety and efficacy of mesenchymal stem cell (MSC) therapy for spinal cord injury (SCI) in humans. Authors synthesized data from 26 studies conducted over the past three decades, organizing information coherently, and addressing methodological aspects such as MSC source, dosage, administration route, and clinical outcomes.
The topic is highly relevant, giving further support to ongoing (and future) clinical translation of MSC based interventions in regenerative medicine. The review integrates both historical and recent evidence, including studies up to 2025, that enhance its scientific value. This contribution is likely to be of significant interest for clinicians, neuroscientists, and researchers in regenerative medicine.
Despite these comments, some concerns need to be addressed. Specifically:
- Define TSM (line 101)
- The terms ASIA and AIS were already defined, please correct (L137)
- Add references to the claims in lines 157 to 160 where necessary.
- Revise grammar in lines 221, 222, 229, 236, 237, 326.
- The term “pathogenetically appropriate procedure” (abstract and conclusion) should be revised to an easier to understand phrase for clarity.
- The claim that MSC transplantation “is safe regardless of the research protocol and severity of injury” should be cautiously stated, as it only appears to be safe across diverse protocols and degrees of injury.
- Some long paragraphs in the Results of Clinical Studies section could be divided for easier reading.
- Please add key information, perhaps modified from the text, to strengthen and further facilitate the understanding of each figure.
- Revise the correct formatting of all references (particularly DOIs).
- Regarding figure 1, I would like to recommend finding a different image for the MRI equipment indicating glial scar reduction, as well as a different image exemplifying neuromuscular conduction as the current one is an example of action potentials. Also, increase the size of the pictures. Additionally, see recommendation 8.
- Regarding figure 2, I would like to recommend enhancing the importance of the message by including information about the molecular mechanism implicated in each of the depicted images. For example, what is delivered when using exosomes? What do they contain? Etc. Also see recommendation 8.
Already provided in the previous section.
Author Response
Rebuttal Letter
Dear Reviewer,
Thank you very much for reviewing the manuscript and for your favorable evaluation of our work. We sincerely appreciate your insightful comments and have implemented the suggested revisions accordingly. We believe that the manuscript in current form meets the requirements for publication.
Please find below our point-by-point response. All revisions in the updated manuscript are marked in blue.
Reviewer: This manuscript presents a comprehensive and well-structured review of clinical studies evaluating the safety and efficacy of mesenchymal stem cell (MSC) therapy for spinal cord injury (SCI) in humans. Authors synthesized data from 26 studies conducted over the past three decades, organizing information coherently, and addressing methodological aspects such as MSC source, dosage, administration route, and clinical outcomes.
The topic is highly relevant, giving further support to ongoing (and future) clinical translation of MSC based interventions in regenerative medicine. The review integrates both historical and recent evidence, including studies up to 2025, that enhance its scientific value. This contribution is likely to be of significant interest for clinicians, neuroscientists, and researchers in regenerative medicine.
We would like to thank you for your time and the accurate review of our manuscript. We have tried to make the following amendments according to your helpful suggestions. Please find our answers and changes below.
Despite these comments, some concerns need to be addressed. Specifically:
1. Define TSM (line 101).
Reply: Thank you! We have removed this abbreviation and abbreviated all references to spinal cord injury as “SCI”.
2. The terms ASIA and AIS were already defined, please correct (L137).
Reply: The correction has been made.
3. Add references to the claims in lines 157 to 160 where necessary.
Reply: Thank you for your suggestion. The listed phrases summarize the general information across all analyzed studies. Based on this, we refer readers to Table 1 for detailed information.
4. Revise grammar in lines 221, 222, 229, 236, 237, 326.
Reply: Done!
5. The term “pathogenetically appropriate procedure” (abstract and conclusion) should be revised to an easier to understand phrase for clarity.
Reply: For the sake of clarity, this term was removed, and both the abstract and conclusion were partially rewritten to better reflect the content of review and improve understanding.
6. The claim that MSC transplantation “is safe regardless of the research protocol and severity of injury” should be cautiously stated, as it only appears to be safe across diverse protocols and degrees of injury.
Reply: Of course! We totally agree with you and have rephrased the sentence accordingly. The only correct conclusion is that MSC transplantation was safe in the studies summarized in our review.
7. Some long paragraphs in the Results of Clinical Studies section could be divided for easier reading.
Reply: Done
8. Please add key information, perhaps modified from the text, to strengthen and further facilitate the understanding of each figure.
Reply: Done
9. Revise the correct formatting of all references (particularly DOIs).
Reply: Done
10. Regarding figure 1, I would like to recommend finding a different image for the MRI equipment indicating glial scar reduction, as well as a different image exemplifying neuromuscular conduction as the current one is an example of action potentials. Also, increase the size of the pictures. Additionally, see recommendation 8.
Reply: Done
11. Regarding figure 2, I would like to recommend enhancing the importance of the message by including information about the molecular mechanism implicated in each of the depicted images. For example, what is delivered when using exosomes? What do they contain? Etc. Also see recommendation 8.
Reply: Following your recommendation, we added a paragraph to the text that explains in detail the mechanisms of action of MSCs, including info about bioactive cargo of MSC-derived exosomes.
Once again we thank you wholeheartedly for thorough review and valuable suggestions. We hope that in its current form the manuscript meets the criteria of MDPI for publishing.
Round 2
Reviewer 2 Report
Comments and Suggestions for Authors
The narrative review format is not suitable for a Q1 journal in this field.
Given the availability of multiple recent systematic reviews and the heterogeneity of MSC studies, only a fully structured systematic review with PRISMA methodology would offer sufficient scientific contribution. The current narrative approach lacks rigor, reproducibility, and critical appraisal, and therefore requires major restructuring before it can be considered.
The paper is written clearly overall, but the English needs some polishing before publication. There are a few grammatical mistakes, repetitions, and awkward sentences that make parts of the text harder to read. I would recommend a careful language check by a native speaker or professional editor to make the manuscript smoother and more consistent.
Author Response
Rebuttle Letter
Reviewer 2. Round 2
The narrative review format is not suitable for a Q1 journal in this field.
Given the availability of multiple recent systematic reviews and the heterogeneity of MSC studies, only a fully structured systematic review with PRISMA methodology would offer sufficient scientific contribution. The current narrative approach lacks rigor, reproducibility, and critical appraisal, and therefore requires major restructuring before it can be considered.
The paper is written clearly overall, but the English needs some polishing before publication. There are a few grammatical mistakes, repetitions, and awkward sentences that make parts of the text harder to read. I would recommend a careful language check by a native speaker or professional editor to make the manuscript smoother and more consistent.
Author’s reply: We appreciate your comments and would like to clarify that several relatively recent systematic reviews on this topic have already been published recently (as it is mentioned in your commentary as well). Since then, results of only a few clinical trials on MSC in SCI have been reported, so converting our narrative review into a systematic one would reduce its novelty. Moreover, many narrative reviews on similar subjects were published in IJMS, indicating that a strictly systematic approach is not a requirement for publication in this journal (or any Q1 journal). Maintaining the narrative format also aligns with our original intent to reach a broader readership, and we would like to keep it as such.
We note that the other three reviewers did not raise concerns about the narrative format and are satisfied with it.
Additionally, although we invited a scientific editing expert who revised the manuscript, you still suggest that there are “a few grammatical mistakes, repetitions, and awkward sentences” and recommend a language check. We kindly ask you to provide specific examples from the revised manuscript to support your claim (this is a common courtesy we extend when serving as reviewers).
Overall, we thank you once again for your input and valuable comments, which have helped us improve our manuscript.
Reviewer 3 Report
Comments and Suggestions for Authors
This updated version shows a remarkable level of enhancement and meets almost all the comments previously raised. The authors have written a well-structured narrative review of the data presented in 26 clinical trials of mesenchymal stem cell (MSC) monotherapy studies involving spinal cord injury (SCI). The discussion and conclusions presented are well-balanced and well-structured. The incorporation of explanations and recent references of 2024-2025 greatly add to the scientific worth of the manuscript. The scope and interest in MSC Monotherapy, not including combination therapy, definitely set the current paper apart from other reviews, and that is a contribution to the field. The paper offers a world perspective of clinical trials, variability of methodology, and a critical discussion of the trends of safety and efficacy.
Minor comments
1. Please explicitly cite Zeng CW et al., Biology 2023 Apr 26; 12(5): 653 (“Multipotent Mesenchymal Stem Cell-Based Therapies for Spinal Cord Injury: Current Progress and Future Prospects”) in the Introduction. Add a short paragraph clarifying how the present review extends beyond this previous work, emphasizing that it uniquely focuses on MSC monotherapy clinical trials (2009-2025) and provides a global, up-to-date synthesis of clinical outcomes rather than mechanistic aspects.
2. It is important that the terms “MSCs transplantation,” “MSC therapy,” and “cell-based therapy” are consistently used. Sticking to a preferred nomenclature will avoid the issue of duplication of terms.
3. If space allows, consider including a concise schematic or timeline summarizing the chronological progression of MSC monotherapy trials (2009-2025). This would visually highlight the global increase in clinical translation and enhance the educational value of the review.
Author Response
Rebuttle Letter
Reviewer 3. Round 2
This updated version shows a remarkable level of enhancement and meets almost all the comments previously raised. The authors have written a well-structured narrative review of the data presented in 26 clinical trials of mesenchymal stem cell (MSC) monotherapy studies involving spinal cord injury (SCI). The discussion and conclusions presented are well-balanced and well-structured. The incorporation of explanations and recent references of 2024-2025 greatly add to the scientific worth of the manuscript. The scope and interest in MSC Monotherapy, not including combination therapy, definitely set the current paper apart from other reviews, and that is a contribution to the field. The paper offers a world perspective of clinical trials, variability of methodology, and a critical discussion of the trends of safety and efficacy.
Minor comments
- Please explicitly cite Zeng CW et al., Biology 2023 Apr 26; 12(5): 653 (“Multipotent Mesenchymal Stem Cell-Based Therapies for Spinal Cord Injury: Current Progress and Future Prospects”) in the Introduction. Add a short paragraph clarifying how the present review extends beyond this previous work, emphasizing that it uniquely focuses on MSC monotherapy clinical trials (2009-2025) and provides a global, up-to-date synthesis of clinical outcomes rather than mechanistic aspects.
Author's reply: Thank you for acknowledging the merits of our work! Yes, it’s a good idea to start this review by introducing previously published most comprehensive works on the subject, such as, indeed, the review by Zeng CW et al, 2023. This will allow us to show how our study stems from and differs from other foundational works.
- It is important that the terms “MSCs transplantation,” “MSC therapy,” and “cell-based therapy” are consistently used. Sticking to a preferred nomenclature will avoid the issue of duplication of terms.
Author's reply: Thank you for your suggestion, we followed your advice and tried to minimize the duplication of terms in the revised text.
- If space allows, consider including a concise schematic or timeline summarizing the chronological progression of MSC monotherapy trials (2009-2025). This would visually highlight the global increase in clinical translation and enhance the educational value of the review.
Author's reply: Great suggestion, we followed your advice and created a timeline (see revised version of the manuscript).
Overall, we thank you once again for your input and valuable comments, which have helped us improve our manuscript. It's been a pleasure to work with you on this text, and your help and guidance are greatly appreciated.
Round 3
Reviewer 2 Report
Comments and Suggestions for Authors
Thank you for your revision. Unfortunately, the major methodological concerns raised in the previous round were not addressed. Since the key issues remain unresolved, the manuscript in its current form does not meet the publication standards of the journal.